# Generating Informative Samples for Risk-Averse Fine-Tuning of Downstream Tasks

**Heasung Kim,** * **Taekyun Lee, Hyeji Kim, and Gustavo de Veciana**
Department of Electrical and Computer Engineering
The University of Texas at Austin
Austin, TX 78712
{heasung.kim, taekyun, hyeji, deveciana}@utexas.edu

## Abstract

Risk-averse modeling is critical in safety-sensitive and high-stakes applications. Conditional Value-at-Risk (CVaR) quantifies such risk by measuring the expected loss in the tail of the loss distribution, and minimizing it provides a principled framework for training robust models. However, direct CVaR minimization remains challenging due to the difficulty of accurately estimating rare, high-loss events—particularly at extreme quantiles. In this work, we propose a novel training framework that synthesizes informative samples for CVaR optimization using score-based generative models. Specifically, we guide a diffusion-based generative model to sample from a reweighted distribution that emphasizes inputs likely to incur high loss under a pretrained reference model. These samples are then incorporated via a loss-weighted importance sampling scheme to reduce noise in stochastic optimization. We establish convergence guarantees and show that the synthesized, high-loss-emphasized dataset substantially contributes to the noise reduction. Empirically, we validate the effectiveness of our approach across multiple settings, including a real-world wireless channel compression task, where our method achieves significant improvements over standard risk minimization strategies.

## 1 Introduction

Risk-averse learning has become increasingly relevant in high-stakes applications where robustness to rare but costly failures is critical. In those domains, models must not only achieve strong average-case performance but also avoid catastrophic errors on atypical inputs. A widely adopted risk measure for capturing such sensitivity is the *Conditional Value-at-Risk* (CVaR), which focuses on the expected loss in the worst-performing $(1 - \beta)$ fraction of the input space (Rockafellar and Uryasev, 2000), making it well-suited for applications requiring robustness guarantees, e.g., large language models, system scheduling, control, medical, wireless communications, and more (Chaudhary et al., 2024, Tan et al., 2017, Ahmadi et al., 2022, Chan et al., 2014, Yang et al., 2022).

Despite its appeal, minimizing CVaR remains challenging in practice. As the quantile level $\beta$ approaches one, loss contributions become dominated by *rare, high-risk inputs that are unlikely to be observed through standard sampling* from the data distribution. Without adequate coverage of these tail events, naive Monte Carlo (MC) methods yield high-variance estimates of CVaR and inefficient optimization, ultimately limiting the reliability of risk-averse training.

---

*Corresponding Author.
Source code: https://github.com/Heasung-Kim/generating-informative-samples-for-risk-averse-fine-tuning-of-downstream-tasks

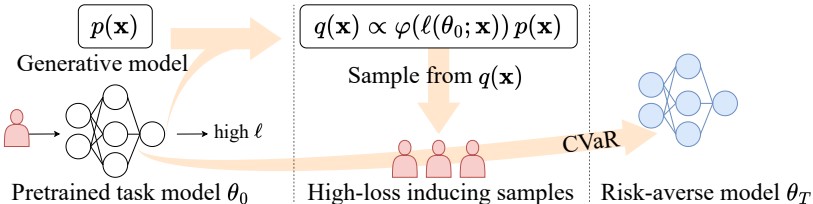

Figure 1: System overview. A score-based generative model is guided using the loss values from a pretrained model to sample high-loss inputs for CVaR optimization.

Recent advances in generative modeling offer new opportunities to address these limitations. In particular, score-based generative models enable expressive and controllable sampling from complex distributions, and have shown promise in tasks ranging from data augmentation to density estimation. Concurrently, pretrained task models are becoming widely available and serve as informative priors for task performance. These developments motivate a fundamental question: *Can we actively generate training inputs that are more informative for CVaR optimization?*

In this work, we propose a novel framework that integrates pretrained (reference) models and generative models to synthesize *informative samples* for risk-averse training. Our key observation is that inputs which induce high loss under a pretrained model are highly beneficial for risk-averse model training. This motivates a data generation strategy that explicitly targets failure modes of the initial model and uses them to guide risk-aware training more effectively.

To realize this idea, we develop a method that uses pretrained loss values to guide a score-based generative model toward a reweighted sampling distribution that emphasizes high-risk inputs. Our approach leverages recent advances in training-free guidance for diffusion models, allowing the generative process to be steered without retraining (Chung et al., 2023; Yu et al., 2023; Kim et al., 2025c). The resulting samples are explicitly biased toward regions where the model is likely to fail and are used to perform CVaR minimization via importance-weighted optimization. The main contributions of this work are summarized as follows.

***Framework Design.*** We propose a novel risk-averse learning framework based on loss-guided generative importance sampling. As illustrated in Figure 1, our approach proceeds in two stages: (i) a pretrained model is used to guide a score-based generative model to sample from a reweighted distribution that emphasizes high-loss inputs; (ii) these samples are then used for CVaR minimization via importance-weighted training, resulting in improved robustness and reduced training noise.

***Theoretical Analysis.*** We provide convergence analysis for our framework and show that, under mild assumptions, generating samples from high-loss regions provably reduces the noise of the CVaR optimization process.

***Empirical Validation.*** We empirically validate our method in both synthetic and real-world settings. In a controlled regression task with highly imbalanced modes, the proposed method successfully synthesizes rare, high-loss samples that are critical for minimizing tail risk. In a real-world application—wireless channel state information (CSI) compression—our method consistently improves CVaR performance in the high $\beta$ regime, compared to existing robust and risk-minimization baselines.

To the best of our knowledge, this is the first work that leverages generative models to perform risk-averse learning by targeting high-loss regions via loss-guided importance sampling.

## 2 Related Work

**Risk-Averse Learning and Conditional Value-at-Risk.** Risk-averse learning seeks to prioritize robustness over average-case performance, especially in high-stakes settings where rare but severe failures are unacceptable. A widely used risk measure in this context is CVaR (Rockafellar and Uryasev, 2000), which quantifies the expected loss in the $(1-\beta)$-worst portion of the input distribution. Due to its ability to explicitly penalize high-loss instances, CVaR has been adopted in a broad range of applications, including finance, credit, operational risk management, robust control in wireless

communications, large language models, and more (Alexander et al., 2006; Andersson et al., 2001; Filippi et al., 2020; Yang et al., 2022; Chaudhary et al., 2024; Chow and Ghavamzadeh, 2014).

However, training with CVaR objective is notoriously challenging due to the high variance in empirical estimates, particularly when targeting extreme quantiles (Troop et al., 2021).

**Importance Sampling for Risk Measures and Optimization.** Importance sampling can be utilized for improving the variance of risk estimation, particularly when rare events dominate the objective. Prior works have explored its use in CVaR estimation and optimization (Bardou et al., 2009; Deo and Murthy, 2021; He et al., 2024a), including sampling-based gradient estimators based on likelihood ratios (Tamar et al., 2015). However, these approaches typically operate on a fixed dataset and focus on reweighting existing samples to reduce estimation variance.

Our method introduces a fundamentally new perspective: we utilize a generative model to directly *synthesize* importance-weighted samples. Under the availability of a generative model, we guide the sample generation process toward high-loss regions using pretrained model losses. This enables us to reduce the noise of CVaR optimization while expanding the effective support of the training distribution.

**Generative Models for Data Augmentation and Downstream Task Learning.** Recent progress in generative modeling, particularly in diffusion and score-based generative models, has enabled high-fidelity sample generation and accurate distribution approximation (Chen et al., 2024; Wang et al., 2024b). These models have been successfully applied across diverse domains for data augmentation, including load forecasting (Xu and Zhu, 2024), medical imaging (He et al., 2024b), and audio synthesis (Bahmei et al., 2022). Recently, generative models are increasingly used to augment training datasets with specific purposes (Zheng et al., 2023), e.g., enhancing semantic diversity (Shivashankar and Miller, 2023; Trabucco et al., 2023), generating label-specific instances (Shao et al., 2019), and bridging distributional gaps between training and test data (Wang et al., 2024a).

**Scope and Distinctiveness of the Proposed Approach.** While prior work has primarily leveraged generative models to enhance generalization by enriching the diversity of training data, our approach adopts a different objective: synthesizing samples that are explicitly *informative for risk-sensitive training*. Rather than uniformly augmenting the training distribution, we concentrate generation toward high-loss regions—those most relevant for CVaR minimization. This targeted generation paradigm aligns directly with risk-averse learning objectives, offering a principled and efficient path toward robust model training.

# 3 Preliminaries and Problem Formulation

Recent advances in generative modeling, particularly score-based generative models, have substantially improved the quality and controllability of synthetic data generation. A key strength of the score-based generative models lies in their ability to support *guided sampling*, where samples can be drawn from a distribution that is shifted or reweighted relative to a base distribution. In this work, we leverage this capability to generate *rare, high-loss-inducing samples* that are underrepresented in standard datasets but critical for risk-sensitive objectives. As we will show, synthesizing such samples provides both theoretical and practical advantages in CVaR minimization.

## 3.1 Score-based Generative Models and Training-Free Guided Sampling

Let $\mathbf{X}^p \in \mathbb{R}^{d_1}$ be a random variable with density $p(\mathbf{x})$, where $\mathbf{x}$ denotes a realization. A generative model seeks to approximate $p(\mathbf{x})$ or its associated score function $\nabla_{\mathbf{x}} \log p(\mathbf{x})$ to enable efficient sampling from the underlying distribution. A key innovation of the recent score-based generative models is to model the score function not directly on $p(\mathbf{x})$, but on noise-perturbed data distributions $p_t(\mathbf{x})$ indexed by a continuous-time parameter $t \in [0, T]$. This enables the data generation process to be formulated as a stochastic differential equation (SDE), which has been shown to enhance both training stability and sample quality (Song et al., 2021). The perturbed distributions are modeled via the Itô SDE:

$$\mathrm{d}\mathbf{X}^p_{(t)} = \mathsf{f}(\mathbf{X}^p_{(t)}, t)\,\mathrm{d}t + \sigma(t)\,\mathrm{d}\mathbf{W}_{(t)}, \quad \mathbf{X}^p_{(0)} \sim p(\mathbf{x}), \tag{1}$$

where $f : \mathbb{R}^{d_1} \times [0, T] \to \mathbb{R}^{d_1}$ is the drift term, $\sigma(t) : [0, T] \to \mathbb{R}$ is the diffusion coefficient, and $\mathbf{W}_{(t)}$ is a standard $d_1$-dimensional Brownian motion. The marginal distribution of $\mathbf{X}^p_{(t)}$ is denoted $p_t(\mathbf{x})$, and the initial distribution $p_0(\mathbf{x})$ corresponds to the data distribution we aim to model.

Given access to the time-indexed score $\nabla_{\mathbf{x}} \log p_t(\mathbf{x})$, samples from $p = p_0$ can be obtained by solving the reverse-time SDE (Anderson, 1982):

$$d\mathbf{X}^p_{(t)} = \left( f(\mathbf{X}^p_{(t)}, t) - \sigma(t)^2 \nabla_{\mathbf{x}} \log p_t(\mathbf{X}^p_{(t)}) \right) dt + \sigma(t) \, d\tilde{\mathbf{W}}_{(t)},$$

where $\tilde{\mathbf{W}}_{(t)}$ denotes reverse-time Brownian motion. Sampling is typically initialized from a simple prior such as a standard Gaussian $p_T(\mathbf{x})$ for large $T$, and trajectories are integrated backward to recover $\mathbf{X}^p_{(0)} \sim p_0$. For notational simplicity, scalar-vector multiplication denotes elementwise scaling.

Beyond sampling from the base distribution $p(\mathbf{x})$, recent advancements in the score-based generative models allow sample generation from modified target distributions of the form $q(\mathbf{x}) \propto w(\mathbf{x})p(\mathbf{x})$, where $w(\mathbf{x})$ is a task-specific importance weight. While classical approaches such as the cross-entropy method (CEM) and fine-tuning of generative models require retraining to realize such reweighted distributions, training-free guidance techniques for the score-based generative models enable approximate sampling from $q$ without modifying the base generative model. These methods exploit the fact that, under the same SDE dynamics in (1), if a process begins from $q(\mathbf{x}) = q_0(\mathbf{x}) \propto w(\mathbf{x})p(\mathbf{x})$ instead of $p$ as $d\mathbf{X}^q_{(t)} = f(\mathbf{X}^q_{(t)}, t) \, dt + \sigma(t) \, d\mathbf{W}_{(t)}$ with $\mathbf{X}^q_{(0)} \sim q = q_0$, then its marginal $q_t$ such that $\mathbf{X}^q_{(t)} \sim q_t$ satisfies $q_t(\mathbf{x}) \propto p_t(\mathbf{x}) \mathbb{E}_{\mathbf{X}^p_{(0)} \sim p(\cdot | \mathbf{X}^p_{(t)} = \mathbf{x})} \left[ w(\mathbf{X}^p_{(0)}) \right]$, where $p(\cdot \mid \mathbf{X}^p_{(t)} = \mathbf{x})$ denotes the conditional distribution of the initial state given state $\mathbf{x}$. Taking the $\log$ and the gradient yields the identity:

$$\nabla_{\mathbf{x}} \log q_t(\mathbf{x}) = \nabla_{\mathbf{x}} \log p_t(\mathbf{x}) + g(\mathbf{x}, t), \quad \text{where} \quad g(\mathbf{x}, t) := \nabla_{\mathbf{x}} \log \mathbb{E}_{\mathbf{X}^p_{(0)} \sim p(\cdot | \mathbf{X}^p_{(t)} = \mathbf{x})} \left[ w(\mathbf{X}^p_{(0)}) \right].$$

This additional term $g$, often referred to as the *guidance*, can be approximated using known quantities such as the score function of $p_t$ and the weight function $w$ (Chung et al., 2023; Kim et al., 2025c; Yu et al., 2023), enabling sampling from approximated $q$ via the reverse-time SDE without any further training.

Motivated by these developments, we propose a new learning framework that leverages guided sample generation to construct *informative training data* specifically tailored for *risk-averse learning*. Rather than drawing training samples uniformly from $p(\mathbf{x})$, we aim to generate samples that contribute to noise reduction in risk-sensitive objectives, thereby improving model robustness to rare but high-loss events.

## 3.2 Risk-Averse Learning via Conditional Value-at-Risk

Our ultimate goal is training of *risk-averse* task models, in which the objective is not merely to optimize expected model performance, but to mitigate the impact of potentially rare but high-loss outcomes. We consider CVaR, one of the most widely adopted risk measures, which builds upon the concept of *Value-at-Risk (VaR)*.

Let $\theta \in \mathbb{R}^{d_2}$ denote the parameters of the task model, and let $\ell(\theta; \mathbf{x})$ be the loss incurred on input $\mathbf{x}$. For a given quantile (confidence) level $\beta \in (0, 1)$, the VaR is defined as the smallest threshold $\alpha$ such that the loss does not exceed $\alpha$ with probability at least $\beta$:

$$\text{VaR}_\beta(\theta) = \min\{\alpha \in \mathbb{R} : \mathbb{P}[\ell(\theta; \mathbf{X}^p) \leq \alpha] \geq \beta\} \tag{2}$$

where $\mathbb{P}[\ell(\theta; \mathbf{X}^p) \leq \alpha] = \int_{\ell(\theta; \mathbf{x}) \leq \alpha} p(\mathbf{x}) \, d\mathbf{x}$, and the distribution is assumed to be continuous with respect to $\alpha$. While VaR captures a quantile of the loss distribution, it does not reflect the magnitude of losses beyond the threshold. The Conditional Value-at-Risk addresses this by computing the expected loss in the tail beyond $\text{VaR}_\beta(\theta)$:

$$\text{CVaR}_\beta(\theta) = \mathbb{E}_{\mathbf{X}^p \sim p}[\ell(\theta; \mathbf{X}^p) \mid \ell(\theta; \mathbf{X}^p) \geq \text{VaR}_\beta(\theta)]. \tag{3}$$

Our objective is to find model parameters $\theta^*$ that minimize the CVaR at a given quantile level $\beta$:

$$\theta^* = \underset{\theta \in \mathbb{R}^{d_2}}{\arg\min} \, \text{CVaR}_\beta(\theta). \tag{4}$$

---

**Algorithm 1** Risk-Averse Model Training via Loss-Guided Importance Sample Generation

---

**Input:** Initial model $\theta_0$, generative model $\nabla \log p_t(\mathbf{x})$, confidence level $\beta$, function $\varphi$, dataset $\mathcal{B}$

**Output:** Risk-averse model $\theta_{K+1}$

1: Generate importance samples $\{\mathbf{x}_i\}_{i=1}^{B_q} \sim q(\mathbf{x}) \propto \varphi(\ell(\theta_0; \mathbf{x})) \, p(\mathbf{x})$
2: Compute $Z = \mathbb{E}_{\mathbf{X}^p \sim p}[\varphi(\ell(\theta_0; \mathbf{X}^p))]$
3: Initialize $\alpha_0$
4: **for** $k = 1$ **to** $K + 1$ **do**
5:     Sample data pair $\{\mathbf{x}, \ell(\theta_0; \mathbf{x})\}$ from $q$
6:     MC estimation of $\alpha_{k-1} + \mathbb{E}_{\mathbf{X}^q \sim q}\left[\frac{Z(\ell(\theta_{k-1}; \mathbf{X}^q) - \alpha_{k-1})^+}{\varphi(\ell(\theta_0; \mathbf{X}^q))(1-\beta)}\right]$
7:     $(\theta_k, \alpha_k)^\top \leftarrow \texttt{SubGradientDescent}(\partial F_\beta, \theta_{k-1}, \alpha_{k-1})$

---

This objective is known to be equivalent to the following unconstrained optimization problem (Rockafellar and Uryasev, 2000) as

$$\theta^*, \alpha^* = \underset{\theta \in \mathbb{R}^{d_2}, \alpha \in \mathbb{R}}{\operatorname{argmin}} F_\beta(\theta, \alpha) \quad \text{where} \quad F_\beta(\theta, \alpha) = \alpha + \frac{1}{1-\beta} \mathbb{E}_{\mathbf{X}^p \sim p}[(\ell(\theta; \mathbf{X}^p) - \alpha)^+]. \quad (5)$$

Here, $(x)^+ = \max(x, 0)$ denotes the positive part function. The solution $\alpha^*$ corresponds to $\text{VaR}_\beta$, and $\theta^*$ minimizes $\text{CVaR}_\beta$.

## 4 Risk-Averse Model Training via Loss-Guided Importance Samples

While the CVaR objective in (5) provides a principled framework for addressing tail-risk scenarios in downstream tasks, its empirical estimation is particularly challenging, especially at high confidence levels ($\beta \to 1$). In this regime, the corresponding VaR threshold $\alpha$ becomes large, and the expectation term $\mathbb{E}_{\mathbf{X}^p \sim p}[(\ell(\theta; \mathbf{X}^p) - \alpha)^+]$ becomes increasingly difficult to estimate due to the *rarity of high-loss instances*, for which $(\ell(\theta; \mathbf{X}^p) - \alpha)^+$ is nonzero. For most samples from $p(\mathbf{x})$, this term evaluates to zero, leading to high variance and poor gradient signals during training. Consequently, naive sampling from the base distribution $p(\mathbf{x})$, even with a generative model, becomes inefficient and often requires an infeasibly large number of samples to stably estimate the CVaR objective for training.

**Key Idea.** To address these, we propose leveraging the generative model to sample from an *importance-weighted* distribution $q(\mathbf{x})$ tailored to highlight high-loss regions. Based on the availability of a pretrained model $\theta_0$ and a score-based generative model capable of sampling from $p(\mathbf{x})$, our approach consists of two components: (i) Sample inputs from a weighted distribution $q(\mathbf{x}) \propto \varphi(\ell(\theta_0; \mathbf{x})) \, p(\mathbf{x})$, where $\varphi : \mathbb{R}_{\geq 0} \to \mathbb{R}_{\geq 0}$ is a nondecreasing function that prioritizes high-loss examples. (ii) Use the importance samples to perform CVaR minimization via importance-weighted MC estimation.

Intuitively, this sample generation strategy concentrates on examples that are informative for CVaR optimization, those in the tail of the loss distribution. These rare, high-loss instances expose the model to critical failure modes and enable more effective risk-averse training. We refer to our approach as *Risk-Averse Model training via loss-guided Importance Samples* (RAMIS).

### 4.1 Algorithm

Algorithm 1 takes as input: an initial pretrained model $\theta_0$, a score-based generative model capable of sampling from $p(\mathbf{x})$, a target quantile level $\beta \in (0, 1)$, and a non-decreasing weighting function $\varphi$ used to construct the importance sampling distribution. We assume access to a dataset $\mathcal{B} = \{\mathbf{x}_i\}_{i=1}^B$, where each $\mathbf{x}_i$ is drawn i.i.d. from the base distribution $p(\mathbf{x})$. This dataset may be externally provided or synthesized via the generative model.

Line 1: We generate $B_q$ samples from the importance-weighted distribution, $q_0(\mathbf{x}) = q(\mathbf{x}) \propto \varphi(\ell(\theta_0; \mathbf{x})) \, p(\mathbf{x})$, by guiding the generative model using loss values computed under $\theta_0$. In score-based generative models, this corresponds to solving the following reverse-time SDE:

$$d\mathbf{X}_{(t)}^q = \left(\mathsf{f}(\mathbf{X}_{(t)}^q, t) - \sigma(t)^2 \nabla_{\mathbf{x}} \log q_t(\mathbf{X}_{(t)}^q)\right) dt + \sigma(t) \, d\tilde{\mathbf{W}}_{(t)}. \quad (6)$$

The specific implementation of this guided importance sampling process may vary based on the chosen generative model guidance method and is detailed in Appendix B.

Line 2: The expectation of the importance weight function $\varphi(\ell(\theta_0; \mathbf{x}))$ over the base distribution $p(\mathbf{x})$ is computed as $\mathbb{E}_{\mathbf{X}^p \sim p}[\varphi(\ell(\theta_0; \mathbf{X}^p))] = Z$. This normalization factor $Z$ can be approximated as $Z \approx \frac{1}{|\mathcal{B}|} \sum_{\mathbf{x} \in \mathcal{B}} \varphi(\ell(\theta_0; \mathbf{x}))$ and is utilized in subsequent optimization iterations.

Lines 4–7: At each training step, we draw a sample from $q(\mathbf{x})$ along with its corresponding importance weight $\varphi(\ell(\theta_0; \mathbf{x}))$. The CVaR objective $\alpha_{k-1} + \mathbb{E}_{\mathbf{X}^q \sim q} \left[ \frac{Z(\ell(\theta_{k-1}; \mathbf{X}^q) - \alpha_{k-1})^+}{\varphi(\ell(\theta_0; \mathbf{X}^q))(1-\beta)} \right]$ is estimated via MC, which corresponds to a variational form in (5). Note that the likelihood ratio $p(\mathbf{x})/q(\mathbf{x})$ simplifies to $Z/\varphi(\ell(\theta_0; \mathbf{x}))$. We perform subgradient-based optimization using a subroutine `SubGradientDescent` (see Appendix A for details), updating both $\theta$ and $\alpha$ in the direction that minimizes the estimated CVaR objective.

**Importance Sampling Mechanism.** The proposed approach differs fundamentally from conventional importance sampling techniques, which re-evaluate model performance at each iteration and dynamically adjust sampling probabilities over a fixed dataset. Such methods introduce additional per-iteration computational overhead (El Hanchi and Stephens, 2020; Needell et al., 2014; Zhao and Zhang, 2015).

In contrast, our framework adopts a fixed importance sampling distribution constructed prior to training. We *guide the generative model to directly produce samples* from the target importance-weighted distribution. This eliminates the need for iterative reweighting or per-batch loss evaluations. Importantly, during training (Lines 4–7 in Algorithm 1), our method introduces *no additional computational overhead* beyond a lightweight scalar reweighting of the loss term $(\ell(\theta; \mathbf{x}) - \alpha)^+$.

## 4.2 Theoretical Analysis

In this subsection, we present a convergence analysis of the proposed RAMIS framework and justify how loss-guided importance sampling improves risk-averse training. Specifically, we show that sampling from a reweighted distribution, which is biased toward high-loss regions under a reference model, reduces the noise of stochastic gradient descent.

**Assumption 1** (Convexity, smoothness, and bounded loss). *For all $\mathbf{x} \in \mathbb{R}^{d_1}$, $\ell(\theta; \mathbf{x})$ are convex, continuously differentiable, $0 < \ell(\theta; \mathbf{x}) < M < \infty$, and $\ell(\theta; \mathbf{x})$ and the norm of $\nabla \ell(\theta; \mathbf{x})$ are $L_1$-smooth and $L_2$-Lipschitz, respectively. For all $k \in [0, K]$, $\|\theta_k\| \leq \kappa$.*

Assumption 1 implies the standard convexity and smoothness of the loss function. We provide a formal definition of convexity and smoothness in Appendix A. Also, the parameterized model norm is bounded. Building on the CVaR minimization analysis of Meng and Gower (2023), which relies on the stochastic model-based framework of Davis and Drusvyatskiy (2019), we have the following convergence property.

**Theorem 1** (Convergence Rate). *Suppose that Assumption 1 holds and over iterations $k = 1, \ldots, K + 1$, Algorithm 1 uses realizations $\mathbf{x}_k$ that are i.i.d. with $q$. Let $\{\phi_k\}$ be the iterates generated by Algorithm 1 such that $\phi_k = (\theta_k, \alpha_k)^\top$, $\phi^*$ is a minimizer of $F_\beta$, and set $\lambda_k = \frac{\lambda}{\sqrt{K+1}}$. For a given quantile $\beta$, we have*

$$\mathbb{E}\left[ F_\beta \left( \frac{1}{K+1} \sum_{k=1}^{K+1} \phi_k \right) - F_\beta(\phi^*) \right] \leq \frac{\|(\theta_0, \alpha_0)^\top - \phi^*\|^2}{2\lambda\sqrt{K+1}} + \frac{\lambda \hat{v}(q)}{\sqrt{K+1}}, \tag{7}$$

*where $\hat{v}(q) = \mathbb{E}_{\mathbf{X}^p \sim p} \left[ \frac{w^*(\mathbf{X}^p)^2}{(1-\beta)^2} \frac{p(\mathbf{X}^p)}{q(\mathbf{X}^p)} + 1 \right]$ and $w^*(\mathbf{x}) = \left( (\sqrt{2L_1 \ell(\theta_0; \mathbf{x})} + 2L_2\kappa))^2 + 1 \right)^{1/2}$.*

**Remark 1** (Loss-dependent Optimization Noise). Theorem 1 establishes an $\mathcal{O}(1/\sqrt{K})$ convergence rate with a stochastic noise term $\hat{v}(q)$ that depends on the initial loss $\ell(\theta_0; \mathbf{x})$. This dependence on the loss value is well-aligned with the standard results in stochastic optimization (Zhao and Zhang, 2015; Davis and Drusvyatskiy, 2019), where the stochastic noise is governed by the *gradient of the loss*. To understand this relationship more precisely, consider the case where the loss function satisfies a Polyak–Łojasiewicz (PL) condition (Karimi et al., 2016). That is, for some $\mu > 0$ and all $\theta$, $2\mu (\ell(\theta_0; \mathbf{x}) - \ell^*) \leq \|\nabla_\theta \ell(\theta_0; \mathbf{x})\|^2$ where $\ell^* = \min_\theta \ell(\theta; \mathbf{x})$. Note that the PL condition holds for several classes of neural networks

(Liu et al., 2019; Zhou and Liang, 2017; Charles and Papailiopoulos, 2018; Hardt and Ma, 2016).
Under $L_1$-smoothness, $\|\nabla_\theta \ell(\theta_0; \mathbf{x})\|^2 \leq 2L_1 \ell(\theta_0; \mathbf{x})$ and we have

$$2\mu\left(\ell(\theta_0; \mathbf{x}) - \ell^*\right) \leq \|\nabla_\theta \ell(\theta_0; \mathbf{x})\|^2 \leq 2L_1 \ell(\theta_0; \mathbf{x}). \tag{8}$$

This chain of inequalities implies that the *high-loss samples contribute proportionately to the norm of the gradient*.

**Remark 2** (Noise Reduction via Importance Sampling). The term $\hat{v}(q)$ in Theorem 1 is minimized when the sampling distribution $q(\mathbf{x})$ is chosen as $q(\mathbf{x}) \propto w^*(\mathbf{x})p(\mathbf{x})$, which depends on the quantities, potentially impractical to compute in real-world scenarios. To circumvent this, we propose using a non-decreasing surrogate weighting function $\varphi : \mathbb{R}_{\geq 0} \to \mathbb{R}_{\geq 0}$ that approximates the behavior of desired importance weights. Specifically, sampling from the distribution $q(\mathbf{x}) \propto \varphi(\ell(\theta_0; \mathbf{x}))\, p(\mathbf{x})$ reduces the term $\hat{v}(q)$ relative to naive sampling from $p(\mathbf{x})$ under the following condition:

$$\hat{v}(p) \geq \hat{v}(q) \iff \mathbb{E}[w^*(\mathbf{X}^p)^2] \geq \mathbb{E}\left[\frac{w^*(\mathbf{X}^p)^2}{\varphi(\ell(\theta_0; \mathbf{X}^p))}\right] \cdot \mathbb{E}[\varphi(\ell(\theta_0; \mathbf{X}^p))]. \tag{9}$$

We observe empirically that simple choices of $\varphi$, such as the square-root or identity mapping, yield strong performance (Sec. 5.1). In summary, the analysis establishes that loss-guided importance sampling based on the pretrained model can reduce the error of CVaR optimization. Rather than merely increasing the sample size, we leverage score-based generative models to synthesize samples from the proposed reweighted distribution, enabling efficient risk-averse training.

# 5 Experiments

**Evaluation Summary.** We evaluate the effectiveness of our proposed framework across both synthetic and real-world tasks. Specifically, we aim to answer the following questions: *(i) Can we generate high-loss-inducing samples using score-based generative models by pretrained models? (ii) Do these samples improve downstream robustness relative to existing robust optimization methods? (iii) Does the approach generalize to high-stakes, real-world applications?*

To this end, we conduct two sets of experiments: Sec. 5.1: We evaluate our method on a controlled regression task over a Gaussian mixture distribution to assess robustness under data heterogeneity and sample scarcity. Sec. 5.2: We apply our method to a real-world wireless channel state information (CSI) compression task, demonstrating its potential practical utility.

**Baselines and Fairness.** We compare against strong risk-sensitive and robust optimization baselines: Stochastic Subgradient Method (SSGM) for CVaR minimization (Meng and Gower, 2023), DORO (Zhai et al., 2021), $\chi^2$-DRO (Namkoong and Duchi, 2016), and standard ERM (i.e., CVaR at $\beta = 0$). All methods start from the same pretrained checkpoint and are trained on the same number of samples from the same generative model; RAMIS uses the identical budget but replaces standard samples with loss-guided (importance) samples. Running SSGM without importance sampling isolates the contribution of our loss-guided sampling scheme, while comparisons to DORO and $\chi^2$-DRO test whether state-of-the-art robust objectives can mitigate tail risk absent our mechanism. ERM serves as a conventional average-risk baseline.

## 5.1 Risk-Averse Regression over Density-Heterogeneous Gaussians

**Task Overview.** We consider a synthetic regression task where inputs $\mathbf{x} \in \mathbb{R}^2$ are drawn from a Gaussian mixture distribution with three components centered at $(-0.6, 0.6)$, $(0.6, 0.6)$, and $(0.0, -0.6)$, each with standard deviation $0.06$ but with unbalanced mixing weights of $0.01$, $0.001$, and $0.989$, respectively (See leftmost subplot in Figure 2). The objective is to predict the second coordinate $\mathbf{x}_{[1]}$ from the first $\mathbf{x}_{[0]}$ using a quadratic regression model trained via risk minimization methods with Mean Squared Error (MSE) loss. This setup can pose a significant challenge for robust learning due to *the extreme imbalance*: standard training on limited number of samples from $p(\mathbf{x})$ yields poor performance on rare but critical components (e.g., the $0.01$ or $0.001$-weight modes), which dominate the tail risk.

To evaluate our method, we adopt a two-phase training strategy. First, we obtain a pretrained (reference) model $\theta_0$ on $10^2$ samples from $p(\mathbf{x})$. Second, we use $\varphi(x) = x^{1/2}$ to construct an

Table 1: CVaR (mean ± std over 100 trials) across Quantile Levels $\beta$ (lower is better).

| $\beta$ | RAMIS (ours) | SSGM | DORO | $\chi^2$-DRO | ERM |
|---|---|---|---|---|---|
| 0.99 | **0.0774 ± 0.0602** | 0.5131 ± 0.6211 | 0.4940 ± 0.4168 | 0.6124 ± 0.5954 | 0.4461 ± 0.4415 |
| 0.95 | **0.0261 ± 0.0156** | 0.1255 ± 0.1406 | 0.1377 ± 0.2146 | 0.1277 ± 0.1012 | 0.1050 ± 0.0963 |
| 0.90 | **0.0172 ± 0.0080** | 0.0604 ± 0.0604 | 0.0694 ± 0.0903 | 0.0766 ± 0.1154 | 0.0559 ± 0.0481 |
| 0.80 | **0.0116 ± 0.0050** | 0.0329 ± 0.0295 | 0.0361 ± 0.0422 | 0.0380 ± 0.0466 | 0.0301 ± 0.0240 |
| 0.50 | **0.0069 ± 0.0024** | 0.0137 ± 0.0106 | 0.0134 ± 0.0102 | 0.0137 ± 0.0115 | 0.0131 ± 0.0096 |

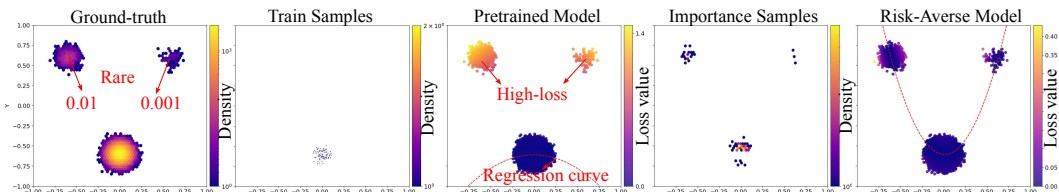

Figure 2: **Visualization of the process.** From left to right: (1) true data distribution $p(\mathbf{x})$, (2) samples drawn from $p(\mathbf{x})$, (3) pretrained model and its loss map, (4) samples drawn from $q(\mathbf{x}) \propto \varphi(\ell(\theta_0; \mathbf{x}))p(\mathbf{x})$, and (5) the final risk-averse model trained on these samples. The loss-guided sampling expands support to rare regions and enables robust optimization.

importance-weighted distribution $q(\mathbf{x}) \propto \varphi(\ell(\theta_0; \mathbf{x})) \, p(\mathbf{x})$ and draw the same number of new samples from this distribution using the corresponding generative model. These samples are then used to train a risk-averse model. More detailed setup and results are provided in Appendix C. Other choices of $\varphi$ are provided in Appendix E.

**Results.** Figure 2 visually illustrates our method. The leftmost panel shows the ground-truth distribution $p(\mathbf{x})$, which includes two rare Gaussian components with low probabilities (0.01 and 0.001). As depicted in the second panel, a limited number of samples drawn from $p(\mathbf{x})$ rarely cover these low-density regions, resulting in limited exposure during training. Consequently, the pretrained model trained on these samples shows high loss in the low-density regions, as reflected in the loss map shown in the third panel–in the colormap, these tail regions appear in yellowish hues, indicating higher loss.

We exploit this loss landscape by constructing an importance-weighted distribution based on $\ell(\theta_0; \mathbf{x})$ and guiding the generative model to sample accordingly. The fourth panel shows samples generated from this reweighted distribution. Despite using the same sample budget, these samples provide *substantially better coverage of the support set, especially in the tails*. The final panel shows that training on these importance samples leads to a risk-averse model that performs reliably across both high-density and tail regions of the input space.

Table 1 reports the CVaR performance across varying quantile levels $\beta$ for our method and baseline approaches. Across all quantile levels, our framework (RAMIS) consistently achieves the lowest CVaR, demonstrating superior robustness in tail-risk regimes. The baseline methods without importance samples exhibit substantially higher risk. These results demonstrate that access to high-loss-inducing importance samples generated via pretrained model guidance provides a distinct advantage that cannot be matched by applying robust optimization techniques over uniformly sampled data.

### 5.1.1 Additional Analysis

To further assess the efficiency of the proposed framework, we analyze (i) the computational cost of generating importance-weighted samples and (ii) the effect of the weighting function $\varphi$, which controls the emphasis placed on high-loss regions. Detailed results and ablation studies are provided in Appendix E.

### 5.2 Risk-Averse Compression of Wireless Channel State Information

**Task Overview.** In wireless communication systems, *Channel State Information* (CSI) captures key physical-layer characteristics such as signal directionality, multipath components, and propagation strength between transmitters and receivers (Lin, 2022). Accurate CSI feedback from the transmitter to the receiver is crucial for tasks like beamforming, scheduling, and adaptive modulation. However, modern CSI matrices are typically high-dimensional, necessitating efficient compression to support

Table 2: CSI Compression CVaR (mean $\pm$ std over 10 trials), in units of $10^{-3}$ (lower is better).

| $\beta$ | RAMIS (ours) | SSGM | DORO | $\chi^2$-DRO | ERM |
|---|---|---|---|---|---|
| 0.99 | **2.2604 $\pm$ 0.0419** | 2.3413 $\pm$ 0.0346 | 2.4987 $\pm$ 0.0744 | 2.5594 $\pm$ 0.0341 | 2.3442 $\pm$ 0.0346 |
| 0.95 | **1.4759 $\pm$ 0.0259** | 1.5260 $\pm$ 0.0198 | 1.6163 $\pm$ 0.0429 | 1.6655 $\pm$ 0.0313 | 1.5273 $\pm$ 0.0198 |
| 0.90 | **1.1292 $\pm$ 0.0189** | 1.1659 $\pm$ 0.0130 | 1.2340 $\pm$ 0.0258 | 1.2697 $\pm$ 0.0225 | 1.1670 $\pm$ 0.0126 |
| 0.80 | **0.7895 $\pm$ 0.0127** | 0.8115 $\pm$ 0.0069 | 0.8576 $\pm$ 0.0100 | 0.8863 $\pm$ 0.0112 | 0.8119 $\pm$ 0.0067 |
| 0.50 | **0.3870 $\pm$ 0.0054** | 0.3954 $\pm$ 0.0024 | 0.4029 $\pm$ 0.0014 | 0.4169 $\pm$ 0.0018 | 0.3953 $\pm$ 0.0024 |
| 0.00 | **0.2030 $\pm$ 0.0027** | 0.2063 $\pm$ 0.0013 | 0.2059 $\pm$ 0.0012 | 0.2061 $\pm$ 0.0014 | 0.2061 $\pm$ 0.0014 |

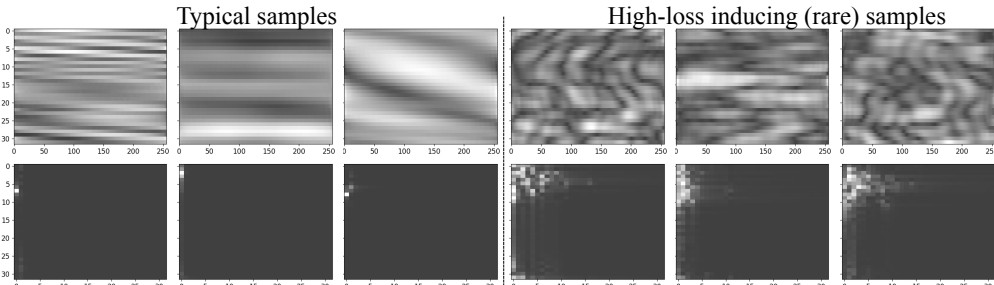

Figure 3: **Visualization of generated samples.** Left three columns: Typical samples with median loss values sampled from the base distribution. Right three columns: High-loss samples generated by pretrained model loss-guided sampling, which exhibit a $6.2 \times 10^{-3}$ reconstruction loss—***rare and unseen*** across $8 \times 10^4$ samples from the base distribution. Top row: Spatial-frequency (Y/X) representation; Bottom row: Angular-delay (Y/X) representation.

bandwidth-constrained channel feedback (Guo et al., 2022). To ensure reliable communication in practical deployments, especially under worst-case scenarios, *risk-averse* compression is essential.

In this experiment, we assess the performance of the proposed method in the context of risk-averse CSI compression. We assume access to a pretrained score-based generative model trained on a CSI dataset generated by the Quadriga simulator (Jaeckel et al., 2021), where a single CSI instance is a $256 \times 32$ complex matrix, and a baseline CSI compressor trained using ERM. The CSI compressor is implemented as a vector-quantized autoencoder (van den Oord et al., 2017), comprising an encoder, a quantization bottleneck, and a decoder. Following Algorithm 1, we guide the pretrained generative model using the *MSE loss values of the initial compressor* to generate informative, high-loss samples. These samples are then used to fine-tune the compressor using a CVaR-based objective. Detailed specifications of the dataset, model architecture, and training parameters are provided in Appendix D.

**Results.** Table 2 reports the CVaR performance in terms of reconstruction distortion (MSE) across various quantile levels $\beta$. Lower distortion indicates better robustness. RAMIS consistently achieves the lowest CVaR in the high-risk regime ($\beta \in \{0.9, 0.95, 0.99\}$), outperforming all baselines, including SSGM, DORO, $\chi^2$-DRO, and ERM. As $\beta$ decreases toward 0, where CVaR approaches the expected loss, the performance gap narrows, and RAMIS converges with SSGM and ERM. This further indicates that RAMIS with importance samples does not degrade the average-case performance, confirming its consistency. The performance gain at the high $\beta$ region supports that the conventional methods, which rely on samples drawn from the original data distribution, are insufficient for minimizing tail risk.

Figure 3 further illustrates the nature of the samples generated via pretrained-loss-guided importance sampling. The top row shows representations in the spatial-frequency domain, while the bottom row visualizes the corresponding angular-delay profiles, computed via 2D inverse FFT (IFFT) with truncation to the low-delay region for interpretability. The left three columns present typical generated samples from the base generative model, chosen as the three median distortion examples by MSE.

By contrast, the right three columns show samples obtained via RAMIS, using the generative model guided by the pretrained model loss. These samples exhibit significantly higher reconstruction distortion, with the average MSE increasing by $4.7 \times 10^{-4}$ and a maximum distortion exceeding $6 \times 10^{-3}$. Notably, the corresponding angular-delay representations reveal more complex scattering patterns, indicating that the proposed framework successfully targets rare, high-loss scenarios that are otherwise underrepresented in the base distribution.

# 6 Discussion, Limitations of Work, and Future Directions

This paper introduces RAMIS, a novel risk-averse learning framework that integrates score-based generative modeling with pretrained model feedback to synthesize high-loss, informative samples for downstream optimization. In contrast to existing generative approaches that aim to increase sample diversity or generalization, our framework targets the *utility* of samples specifically for minimizing tail-risk objectives such as CVaR. By leveraging pretrained loss signals as importance guidance, we enable generative models to contribute directly to risk-sensitive training.

Our study primarily focuses on the theoretical motivation behind loss-guided generative sampling and demonstrates its effectiveness through controlled synthetic experiments and a domain-specific application in wireless communication. While these results validate the core principles of RAMIS, broader applications remain to be explored. As diffusion-based generative models continue to evolve and become increasingly accessible across domains where risk-sensitive optimization is critical, we expect RAMIS to generalize naturally to these settings.

## Acknowledgment

This material is based upon work supported by the National Science Foundation under Grant Nos. 2148224 and 2443857, the ARO Award W911NF2310062, and the ONR Award N000142412542, and is supported in part by funds from OUSD R&E, NIST, and industry partners as specified in the Resilient & Intelligent NextG Systems (RINGS) program and the WNCG/6G@UT.

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
