# OpenReview forum: "Generating Informative Samples for Risk-Averse Fine-Tuning of Downstream Tasks"
_NeurIPS.cc/2025/Conference — NeurIPS 2025 spotlight_

### Official Review · Reviewer_uMpj · 2025-06-30

**Clarity:** 3
**Significance:** 3
**Originality:** 4
**Rating:** 5
**Confidence:** 4

**Summary:**

The paper proposes to use generative models to sample high risk samples for efficient distributionally robust learning. The authors design an appropriate algorithm that takes advantage of guided generative models to do some form of "online importance sampling". The authors present convergence guarantees of their approach. They illustrate the relevance of their approach on synthetic and real datasets.

**Questions:**

- See first weakness point. Can the authors explain why the samples are guided using only the initial point? Why aren't samples refined with new predictors? Please consider reading some of the references given in that part. I sincerely believe that there was a slight misunderstanding on what distribution really matters, but please correct me if I'm wrong. Maybe what the authors missed to highlight is that their procedure should only be used after a first round of empirical risk minimization?
- Can the authors prove the reduced variance they hope their method can bring (on both losses and subgradients)?
- Can the authors explain how they chose the other methods they compared to?
- Can the authors try experiments using extremile rather than the CVAR?

**Ethical Concerns:**

["NO or VERY MINOR ethics concerns only"]

**Final Justification:**

The idea is original and can really open up fruitful avenues for both theory and practice.
The theoretical part is somewhat lacking but it will be a great opportunity for other researchers.
The authors fully addressed all my concerns in a very detailed way. With the hope that they will incorporate their answers to their final version I raised my score to acceptance.

**Limitations:**

See weakness part.

**Quality:**

3

**Strengths And Weaknesses:**

**Strengths**:
- The idea is original, interesting, and can open fruitful avenues both in theory and in practice.
- The empirical illustrations are well crafted.

**Weaknesses**:
- The distribution of the losses depends not only on the inputs but also on the predictor. So generating adverse samples from the distribution of the losses at the initial point, does not necessarily reflect the adequate tail distribution of the losses at the predictor we want to use. Consider taking $\theta_0 = 0$, would the corresponding guided samples be relevant? Probably not. If $\theta_0$ was the optimum of the ERM, then the guided samples would probably be relevant but this highlights the problem: the distribution of losses one wants the tails of depend on the predictor. Said differently, distributionally robust learning is fundamentally a min-max problem, i.e., best predictor over worst possible distribution around training distribution. What the authors have is essentially a single "max" step i.e., a single step towards getting a "hard" distribution from the initial point, but this should be iterated. An effective algorithm would probably consist in iterating the current proposal several times. Weirdly the authors argue in favor of this "fixed importance sampling distribution", which I don't understand. An ideal way to frame the contribution will probably to fully write the original problem in terms of distributions, see Preliminaries in [1] for example.
- The key part of the algorithm (and where it changes from usual distributionally robust optimization (DRO)) is in having access to low variance unbiased estimates of the loss and the subgradients of the CVAR objective in $\alpha$, $\theta$ (so the MC step in the algorithm). The authors do not show that key point. Their argument goes through proving the convergence of the algorithm by some means and with some bounds from which they argue what should be the choice of initial distribution. This circumvolved path hides whether the claimed objective of the authors "getting low variance oracles for optimization" is reached or not.
- CVAR is not the only risk-sensitive measure that could be used. Spectral risk measures like the extremile [2] partially avoid the problem of CVARs thresholding (see also [3] for optimization of spectral risk measures). That said adequate estimation of tails are always at the core of distributionally robust learning even if some other risk measure than the CVAR is used.
- The overall idea hinges upon the assumption that the generative model is sufficiently good to generate high risk data points when such high risk data points are not present in the data. In other words it hinges upon the generalization capabilities of the generative model. It would be best to formalize this, i.e., what generalization capabilities are assumed on the generative model for the overall approach to work. Once again formalizing the problem as in [1] could help.
- Assumptions on Theorem 1 are rather stringent (assuming bounded loss for risk-sensitive applications is always a bit counter intuitive and having other research works making the same assumptions is not a good reason for this). That said, as long as the theoretical results are meaningful to understand (i) the relevance of the approach, (ii) how to further refine e.g. parameters, that's fine. As said above, it would be best for the results to show e.g. lower variance of the stochastic estimates.
- Eq; 8: the bound on square gradient norms does not give a bound on the variance. So the whole argument does not seem sound.
- While the illustrations are very well done, the choice of competing algorithms is not very well justified. Namely, the given contribution could be seen as some form of importance sampling (though not refined over iterates). It would be interesting to compare to such approaches.  For example, [1] provided for example analyses of DRO in online learning setup that would be relevant here.

[1] Levy, Daniel, et al. "Large-scale methods for distributionally robust optimization." Advances in Neural Information Processing Systems 33 (2020): 8847-8860.
[2] Daouia, Abdelaati, Irene Gijbels, and Gilles Stupfler. "Extremile regression." Journal of the American Statistical Association 117.539 (2022): 1579-1586.
[3] Mehta, Ronak, et al. "Distributionally Robust Optimization with Bias and Variance Reduction." The Twelfth International Conference on Learning Representations.

---

> ### Author Rebuttal · Authors · 2025-07-30
>
> We sincerely thank the reviewer for the thoughtful and constructive feedback. For clarity, we refer to individual points using **Q** (Questions) and **W** (Weaknesses), e.g., Q1 refers to the first bullet under _Questions_, W1 to the first under _Weaknesses_.
>
> ---
> **[R3Q1]**  (Q1, W1) . Consider taking $\theta_0 = 0$... Can the authors explain why the samples are guided using only the initial point? Why aren't samples refined with new predictors? /... Maybe what the authors missed to highlight is that their procedure should only be used after a first round of empirical risk minimization?
>
> **[R3A1]**  Thank you for raising this point. We have addressed this question in conjunction with a related question from Reviewer dRNj in **[R2Q2]** and its corresponding response **[R2A2]**. Kindly refer to **[R2A2]** for our detailed explanation.
>
> ---
>
> **[R3Q2]**  (Q2, W6, W2, W5) Eq; 8: the bound on square gradient norms does not give a bound on the variance... / Can the authors prove the reduced variance they hope their method can bring (on both losses and subgradients)? /  The key part of the algorithm is in having access to low variance unbiased estimates of the loss and the subgradients of the CVAR objective ... Their argument goes through proving the convergence of the algorithm by some means and with some bounds from which they argue what should be the choice of initial distribution. This circumvolved path hides whether the claimed objective of the authors "getting low variance oracles for optimization" is reached or not.... / ... it would be best for the results to show e.g. lower variance of the stochastic estimates.
>
> **[R3A2]** We appreciate the reviewer’s insightful comments.
>
> First, we clarify that our method targets reduction of the overall stochastic noise upper bound $\hat{v}(q)$, rather than explicitly minimizing the variance of per-iteration subgradients or per-iteration loss estimates. Our primary goal of designing importance sampling distribution is to minimize the distance between the final trained parameter and the true optimum, as stated in Theorem 1.
>
> Regarding (8), it is introduced to justify our use of an importance sampling distribution proportional to the pretrained model's _loss_. While classical variance reduction analyses for SGD often optimize sampling proportional to the gradient norm squared (Zhao & Zhang, 2015), Eq. (8) shows that the gradient norm is bounded both above and below by a linear function of the loss under the PL condition. This supports our surrogate choice of the pretrained model loss as a practical proxy for gradient magnitude. In other words, *we do not claim that we minimize the gradient squared norm's upper bound*, instead, Eq (8) shows the proportional relationship between the gradient norm and the loss value.
>
> The reason we use the term “variance” following Eq. (8) is that, by taking expectation over the inequality, we obtain a second-order moment of the gradient norm. And the expectation of the pretrained model’s loss appears in $\hat{v}(q)$ with the optimal importance weight function $w*$ (substitution of $w^*$ into $\hat{v}(q)$). We agree that referring to this as “variance” is misleading, and we will revise the statement at line 230 to: _“high-loss samples contribute proportionately to the squared norm of the gradient.”_
>
> That said, we fully agree that demonstrating subgradient variance reduction is important. To fully address the reviewer's concern, we provide additional analysis regarding the subgradient and loss. Theorem 1 and Appendix Eq. (20), inherently address this. The subgradient of the $k$-th iteration objective is denoted $g_k$, and its squared norm is bounded as:
>  $\Vert g_k \Vert^2 \le v(x)^2$ where  $v(x) = \sqrt{ \frac{1}{(1-\beta)^2}\Vert \frac{p(x_k)\partial \ell (\theta_k;x_k)}{q(x_k)} \Vert^2 + 1 +\frac{p(x_k)^2}{q(x_k)^2 (1-\beta)^2} }$. Taking expectation over $q$, on the left hand side, we have the second moment of the subgradient, $\mathbb{E}[\Vert g_k \Vert^2]$ . In (23)-(27), (line 903-906, in Appendix), it directly shows that the stochastic noise $\hat{v}(q)$ is an upper bound of the  $\mathbb{E}[\Vert g_k \Vert^2]$ and  $\hat{v}(q)$ is minimized by the importance sampling distribution with weight $w^*$, as stated in Theorem 1.
>
> For per-iteration loss variance reduction, we consider a $\theta_k$, the model parameter at the $k$-th iteration and corresponding dual form objective is given as $F_{\beta}(\theta_k, \alpha_k) = \alpha_k + \frac{1}{1-\beta}\mathbb{E}[(\frac{p(x)}{q(x)}(\ell(\theta_k;x)-\alpha_k)^{+})]$. Then it is straightforward to see that the variance of this estimate is reduced if: $\mathbb{E}[(\frac{p(x)}{q(x)})((\ell(\theta_k;x)-\alpha_k)^{+})^2] < \mathbb{E}[((\ell(\theta_k;x)-\alpha_k)^{+})^2]$. Considering the pretrained model loss-based importance sampling distribution $q(x) \propto \varphi(\ell(\theta_0;x))p(x)$, we have $\mathbb{E}[\frac{Z((\ell(\theta_k;x)-\alpha_k)^{+})^2}{\varphi(\ell(\theta_0;x))}] < \mathbb{E}[((\ell(\theta_k;x)-\alpha_k)^{+})^2]$ where $Z$ is the normalizing constant. (Informal statement: Intuitively, the result suggests that if instances with high loss under $\theta_0$ continue to have high loss during fine-tuning (i.e., $\theta_k$), then the importance sampling concentrates on high-loss regions, thereby reducing variance. As discussed in [R2A3], this assumption can be reasonable in the context of fine-tuning from a well-initialized model, and high-loss-inducing samples often require multiple optimization steps to be effectively addressed in risk-averse training.)
>
> We will incorporate this discussion and explanation into the revised manuscript. Thank you again for your constructive suggestion.
>
>
> ---
>
> **[R3Q3]**  (Q3, W7) While the illustrations are very well done, the choice of competing algorithms is not very well justified....  / Can the authors explain how they chose the other methods they compared to?
>
> **[R3A3]**
> Thank you for this question. In our results table, SGM uses the same configuration but without importance sampling. This isolates the contribution of our pretrained-model loss guided importance sampling scheme and demonstrates the benefit. We selected DORO and $\chi^2$-DRO to assess whether state-of-the-art robust optimization methods can address the tail-risk challenges without our proposed mechanism- i.e., importance samples. The results highlight that such methods do not inherently resolve the difficulties addressed by RAMIS, underscoring the need for our new approach. Finally, ERM is included as a conventional baseline to illustrate the limitations of average-risk optimization in high-risk scenarios.
>
> We will clarify this motivation in the revised version.
>
>
>
> ---
>
> **[R3Q4]** -   (W4) The overall idea hinges upon the assumption that the generative model is sufficiently good to generate high-risk data points when such high-risk data points are not present in the data. It would be best to formalize this, i.e., what generalization capabilities are assumed on the generative model for the overall approach to work...
>
> **[R3A4]**  As the reviewer notes, our framework assumes that the generative model can sample from the true distribution $p(x)$, including high-risk regions. This is feasible under our setup as we use a score-based generative model, where the input to our importance sampler is the time-dependent score $\nabla \log p_t(x)$. This implies that the model can generate samples following $p(x)$.
>
>  We will formalize this assumption more clearly in the revised manuscript.
>
> ---
>
> **[R3Q5]** (Q4, W3) CVAR is not the only risk-sensitive measure that could be used.... adequate estimation of tails are always at the core of distributionally robust learning even if some other risk measure than the CVAR is used / Can the authors try experiments using extremile rather than the CVAR?
> **[R3A5]** We appreciate the reviewer's suggestion. Our framework is indeed general and can be applied to other risk measures.
>
> Following the Extremile formulation in [2] and the notations, we use: various $\tau$ values raning from 0.5 to 0.99, $r(\tau)=\log(1/2)/\log(\tau)$, $K_{\tau}(t) = t^{r(\tau)}$, and $J_{\tau}(F(x))$ which is scaled by the squared loss. We compare *RAMIS + Extremile* (our framework applied to Extremile with importance sampling) against naive Extremile optimization under the same experimental setup as in Figure 2 of the paper.
>
> | $\tau$    | RAMIS + Extremile (mean ± std) | Extremile (mean ± std) |
> |------|---------------------------|------------------------|
> | 0.99 | **0.187 ± 0.144**             | 0.339 ± 0.207          |
> | 0.95 | **0.064 ± 0.021**             | 0.090 ± 0.044          |
> | 0.90 | **0.044 ± 0.015**             | 0.048 ± 0.021          |
> | 0.80 | **0.0288 ± 0.0062**           | 0.0289 ± 0.0041        |
> | 0.50 | **0.0158 ± 0.0015**           | 0.0179 ± 0.0025        |
>
> In all settings, RAMIS with importance sampling outperforms the naive baseline, confirming both the method’s flexibility and effectiveness across different risk measures. We will include these new results in the revised manuscript.
>
>
>
>
> ---
> We appreciate the reviewer’s valuable feedback again. We believe we have addressed all concerns; if so, we kindly ask the reviewer to consider a higher score. We are happy to provide additional clarification if needed.

---

> > ### Author Response · Authors · 2025-08-02
> >
> > We would like to clarify that **[R3A1]** refers to **[R2Q3]/[R2A3]**, which directly addresses the points raised in both **[R3Q1]** and **[R2Q3]**. We kindly invite the reviewer to refer to that response for the detailed explanation. Thank you again.

---

> > > ### Author Response · Authors · 2025-08-06
> > >
> > > Dear Reviewer,
> > >
> > > Thank you again for your insightful and constructive feedback. We hope our responses have addressed your concerns—particularly regarding the use of a fixed importance sampling distribution, the extensibility of our method through iterative application, the reduction of subgradient and loss variance, the choice of baselines, the capability of the generative model, and the applicability of our framework to alternative risk measures.
> > >
> > > Please let us know if any further clarification of our responses to your comments on these points would be helpful.
> > >
> > > Best regards,
> > >
> > > the authors

---

> > > > ### Comment · Reviewer_uMpj · 2025-08-06
> > > >
> > > > I beg your pardon for the delay, I had less good internet connection than I hoped.
> > > >
> > > > The answers of the authors to my concerns are clear and completely address them.
> > > > Again, the idea is original, interesting, and can open fruitful avenues both in theory and in practice. The current theory of the paper may be slightly limited but this should not overshadow the contribution. In particular I hope the authors will insist on the "finetuning" setup as well as the potential for iterative RAMIS as explained in R2A2.
> > > > I will therefore raise my score. I hope generally that the authors will take time to incorporate the answers to the varous concerns brought up by the reviewers.
> > > >
> > > > Thank you for the detailed answers!

---

> > > > > ### Author Response · Authors · 2025-08-09
> > > > >
> > > > > We sincerely thank you again for reviewing our work and for your valuable feedback. We will incorporate the additional clarifications on the fine-tuning setup and iterative RAMIS, along with the new results, in the revised manuscript.

---

### Official Review · Reviewer_dRNj · 2025-07-07

**Clarity:** 3
**Significance:** 3
**Originality:** 3
**Rating:** 4
**Confidence:** 3

**Summary:**

This paper develops an interesting approach to using guided generative models to produce synthetic data samples for more robust optimization of the conditional value-at-risk (CVaR). Training for good CVaR is challenging, especially when the target risk quantile is quite high. At these extreme quantiles the events that we would like to produce low-loss predictions for are very rare, and therefore relying on standard datasets to measure and optimize empirical estimates of the CVaR leads to very high variance. This work leverages generative models with guidance to (1) sample from a distribution $q(x)$ that is tilted towards high-loss samples, i.e., $q(x) \propto \phi(\ell(\theta_0; x))p(x)$, and then (2) use importance weighting on samples from $q$ to minimize a MC estimate of the CVaR. The idea takes advantage of existing ideas in generative modeling and importance weighted optimization, but is elegantly designed and well-motivated.

**Questions:**

- I'm curious about the effect that the method has on the standard risk. In Table 2 we start to see ERM improve over RAMIS for lower values of $\beta$. While its expected that methods that optimize the CVaR will not also optimize the standard empirical risk, it would be nice to show these results in the table as well --- even if just to understand the tradeoffs.

- I understand that the fixed importance sampling distribution is computationally efficient, but am curious about how sensitive this is to the choice of the initial model $\theta_0$. For example, if $\theta_0$ is uniformly bad, it will not be informative. Is there any other reason _not_ to iterative update the reference model used for loss guidance?

Minor comments:
- L256 / L291 / Table 1 acronym confusion: Do you use the stochastic subgradient method, the stochastic prox-linear method, or the SPL+ method of the cited Meng & Gower? (And if the latter, please specify a different acronym that SGM, as SGM is already widely used in the paper).

**Ethical Concerns:**

["NO or VERY MINOR ethics concerns only"]

**Final Justification:**

I do believe the contribution is good and well presented as written above. Still, the evaluation is limited as the authors have acknowledged.

**Limitations:**

Yes

**Quality:**

3

**Strengths And Weaknesses:**

As described above, I think that the main idea in the paper makes sense, is well described, and forms a nice connection between recent advances in generative modeling with guidance and data augmentation techniques for risk-averse optimization. The convergence analysis and the connection to importance weighting reducing the variance of the stochastic gradient update provide further support and justification for this approach (which are aligned with standard results in the literature). That said, the empirical results are a bit weak (as I believe the authors already somewhat acknowledged in their discussion of limitations and future directions). Specifically, the paper evaluates on one synthetic task and just one real data task on wireless communications. It would be more compelling to see results on tasks where generating high quality samples may be more challenging --- such as in image generation or text sequences (e.g., to name a few suggestions: optimizing CVaR for image classification in ImageNet, radiology report generation on MIMIC-CXR, or reward modeling / preference learning on language model responses).

---

> ### Author Rebuttal · Authors · 2025-07-30
>
> **[R2Q1]** -   ...the effect that the method has on the standard risk. In Table 2 we start to see ERM improve over RAMIS for lower values of $\beta$. While its expected that methods that optimize the CVaR will not also optimize the standard empirical risk, it would be nice to show these results in the table as well --- even if just to understand the tradeoffs.
>
> **[R2A1]** We thank the reviewer for the insightful comment.
>
> In Table 2, each row corresponds to a different value of $\beta$. As $\beta$ decreases, the CVaR objective gradually approaches the standard ERM. In fact, when $\beta = 0$, CVaR becomes equivalent to ERM, and RAMIS can be directly used for minimizing the standard empirical risk. To fully address the reviewer’s suggestion, we conduct two complementary analyses.
>
> **(1) Standard ERM performance with $\beta = 0$:**
> First, based on the same configuration corresponding to Table 2, we set $\beta = 0$ for the objective and all methods under the standard ERM. Below table shows the results (in unit of $10^{-3}$).
> |   | RAMIS (ours)     | SGM             | DORO             | χ²-DRO           | ERM              |
> |-------|------------------|------------------|------------------|------------------|------------------|
> | $\beta=0$  | 0.2032  | 0.2032  | 0.2029  | 0.2029   | 0.2028  |
>
> In this case, the CVaR objective function is equivalent to ERM. As they are minimizing the same objective function, their performance is nearly identical, as expected.
>
> (2) Evaluating trained models at different non-target quantile levels:
>
> Next, we fix $\beta = 0.99$ (1% tail-risk focus) and evaluate the resulting models **across a range of quantile levels** $\beta$
> | β     | RAMIS  | SGM             | DORO             | χ²-DRO           | ERM              |
> |-------|------------------|------------------|------------------|------------------|------------------|
> | 0.99 (true target) | **2.10 ± 0.0028**    | 2.21 ± 0.0341    | 2.28 ± 0.0032    | 2.50 ± 0.2330    | 2.22 ± 0.0016    |
> | 0.95  | **1.41 ± 0.0283**    | 1.43 ± 0.0153  | 1.49 ± 0.0034  | 1.72 ± 0.2293  | 1.44 ± 0.0019  |
> | 0.90  | **1.09 ± 0.0121**    | 1.10 ± 0.0059  | 1.14 ± 0.0029  | 1.38 ± 0.2267  | 1.10 ± 0.0018  |
> | 0.80  | 0.77 ± 0.0028    | 0.78 ± 0.0012  | 0.80 ± 0.0014  | 1.02 ± 0.2249  | **0.76 ± 0.0013**  |
> | 0.50  | 0.39 ± 0.0069    | 0.41 ± 0.0038  | 0.43 ± 0.0004  | 0.63 ± 0.2251  | **0.38 ± 0.0003**  |
> | 0.00  | 0.21 ± 0.0048    | 0.23 ± 0.0045  | 0.24 ± 0.0005  | 0.44 ± 0.2229  | **0.20 ± 0.0001**  |
>
> These results highlight the expected tradeoff: RAMIS, the proposed method, achieves superior performance in target high-risk regimes (target $\beta=0.99$ and $\beta=0.95, \beta=0.9$), while ERM excels in average-case scenarios. This supports the intuition that optimizing for CVaR enhances robustness to rare, high-loss inducing samples, at the potential cost of standard performance.
>
> We will include these results and discussions in the revised manuscript. We thank the reviewer again for raising this important point.
>
> ---
> **[R2Q2]** - The empirical results are a bit weak (as I believe the authors already somewhat acknowledged in their discussion of limitations and future directions)...
>
> **[R2A2]** Our synthetic experiment was intentionally designed as a controlled setting to rigorously evaluate the core mechanism of RAMIS, whether our framework can identify and generate high-loss inducing samples that are often missed by standard sampling, by using a pretrained initial model. By constructing a Gaussian mixture with extreme mode imbalance (Fig. 2), we create a challenging yet interpretable testbed to directly visualize sample coverage and assess risk sensitivity.
>
> We evaluate RAMIS on wireless CSI compression to demonstrate its applicability to high-dimensional, practical data where compression robustness to rare but high-distortion channel conditions is essential. This task exemplifies a real-world setting where standard training underrepresents such cases, and improved risk sensitivity directly translates to better signal reconstruction under challenging conditions.
>
> That said, we agree that applying RAMIS to broader domains is an important next step, and we plan to explore such extensions in future work.
>
> ---
> **[R2Q3]** -   I understand that the fixed importance sampling distribution is computationally efficient, but am curious about how sensitive this is to the choice of the initial model. For example, if is uniformly bad, it will not be informative. Is there any other reason not to iterative update the reference model used for loss guidance?
>
> **[R2A3]**   Thank you for this question.
>
> [Fixed Importance Sampling Distribution] As the reviewer noted, a primary motivation for using a fixed importance sampling distribution is **computational efficiency**: it avoids repeated loss evaluations, recomputation of the normalization constant, and allows reuse of generated samples. Iteratively updating the sampling distribution per optimization step introduces significant overhead, as highlighted in prior literature (e.g., Zhao & Zhang, 2015). Rather than performing per-step updates, our practical importance sampling distribution design focuses on minimizing the stochastic noise bound, i.e., the final expected model gap.
>
> [Choice of $\theta_0$]
> Moreover, our framework is designed for the **fine-tuning** setting, assuming access to a reasonably informative pretrained model $\theta_0$, as reflected in the title of this work. Pretrained models for many tasks are now widely available and can serve as strong baselines and provide informative guidance for importance sampling. In typical fine-tuning scenarios, only a small number of additional training iterations are required compared to training from scratch. Empirically, we demonstrate that using a fixed importance sampling distribution in this regime yields superior performance compared to existing robust optimization baselines.
>
>  If a pretrained model $\theta_0$ assigns uniformly low or high losses, it indicates the absence of a meaningful loss tail—i.e., a scenario where risk-sensitive training is unnecessary.
>
> Using an untrained or randomly initialized model (e.g., $\theta_0 = 0$) leads to uninformative importance weights, since the loss-induced distribution fails to highlight relevant failure modes. *Such scenarios are outside the scope of our intended use case*, which targets the increasingly common setting where pretrained models are available and serve as reasonable reference surrogates.
>
> [Iterative Application]
> That said, **RAMIS is fully compatible with iterative refinement**. Specifically, Algorithm 1 can be applied recursively: the model $\theta_K$ obtained from one round can serve as the new reference model for generating the next round of guided samples. Our theoretical analysis remains valid under this substitution, and the bound in Eq. (7) becomes strictly tighter due to improved reference model quality.
>
> In our experiments, the fixed pretrained-model loss guided importance sampling distributions already achieve strong CVaR performance. However, when starting from an untrained model that requires many iterations to converge, repeated application of RAMIS will lead to improved performance.
>
>
> We will clarify these points in the final version of the manuscript and explicitly state that RAMIS supports iterative application when appropriate.
>
>
> [Ref-R2-1] Zhao, Peilin, and Tong Zhang. "Stochastic optimization with importance sampling for regularized loss minimization." _International Conference on machine learning_. PMLR, 2015.
>
> ---
> **[R2Q4]**-   L256 / L291 / Table 1 acronym confusion
>
> **[R2A4]** Thank you for the careful proofreading. In L256 and Table 1, "SGM" refers to the Stochastic Subgradient Method, where SPL is a typo, which should be SGM. We will revise the manuscript to remove redundant or conflicting abbreviations.
>
>
> ---
> We appreciate the reviewer’s valuable feedback again. We believe the revised discussion addresses all the concerns, and we kindly ask the reviewer to consider a higher score. We are happy to provide additional clarification if needed.

---

> > ### Author Response · Authors · 2025-08-06
> >
> > Dear Reviewer,
> >
> > Thank you again for your thoughtful and constructive feedback. We hope our responses have addressed your comments—particularly regarding the tradeoffs between CVaR and standard risk, the scope of our empirical evaluation, and the use of a fixed importance sampling distribution, as well as the extensibility of our method through iterative application.
> >
> > Please do let us know if we could provide any further clarifications to our response to your comments on these points.
> >
> > Best regards,
> >
> > the authors

---

> > > ### Comment · Reviewer_dRNj · 2025-08-07
> > >
> > > Thank you for the answers to my questions. I will maintain my positive score, and look forward to future results on applying and verifying the performance of RAMIS on broader domains.

---

> > > > ### Author Response · Authors · 2025-08-09
> > > >
> > > > We sincerely appreciate your insightful review and constructive feedback once again. We look forward to exploring broader applications of our method in future work, and we will incorporate all additional results into the revised manuscript.

---

### Official Review · Reviewer_hgR6 · 2025-07-10

**Clarity:** 3
**Significance:** 3
**Originality:** 3
**Rating:** 5
**Confidence:** 3

**Summary:**

The paper provides a new approach for optimizing conditional value-at-risk (CVaR) over distributions generated from diffusion-based generative models. The primary challenge of optimizing CVaR is typically due to being able to efficiently sample high loss samples in order to reduce the variability of CVaR objective. To address the challenge, the paper's approach proposes an algorithm for reweighting the sampling distribution of the generative diffusion models and theoretically analyze the convergence of their approach. They conclude with numerical experiments that compare their approach with benchmark approaches on the same sampling budget and show their approach better optimizes CVaR.

**Questions:**

1. Can any comparison be made about the theoretical convergence rate between the proposed method and other benchmark methods? For example, how does the rate presented in this paper compare to the rates SGM, DORO, etc.? Similarly, how does the analysis differ from previous works that also use importance sampling for non-generative models?

2. It's not clear to me how to select $\phi$, are there empirical results that compare different choices of $\phi$?

**Ethical Concerns:**

["NO or VERY MINOR ethics concerns only"]

**Final Justification:**

After reviewing the author's rebuttal which cleared up my remaining questions and the reviews of other reviews, I will maintain my score of 5. While the numerics may be limited, the approach seems straightforward and effective which warrants an acceptance.

**Limitations:**

Yes

**Paper Formatting Concerns:**

The appendix could be included in the pdf rather than the supplementary files zip.

**Quality:**

3

**Strengths And Weaknesses:**

Strengths
1. The paper provides a straightforward and effective method for improving the efficiency of generating accurate estimates of CVaR. The approach highlighted by Algorithm 1 and corresponding descriptions for each line were clear and easy to follow.
2. The paper's theoretical analysis seems correct and provides good motivation for why their approach is effective and what inputs affect the convergence rate. The provide remarks are useful in highlighting challenges in their approach and heuristic solutions for addressing the challenges.
3. The work feels relevant and important given recent works in generative models. Given the complexity of such models and their outputs, it makes sense that obtaining sufficient samples to optimize metrics like CVaR would be costly. Moreover, given the prevalance of these models, it makes sense that practitioners would be interested in training models that optimize the CVaR of outcomes that leverage these models.
4. The numerical section while simple highlights intuition for why the proposed approach is effective. Leveraging real-world data also highlights the practical effectiveness of their method.

Weaknesses
1. The paper does not compare or comment on their theoretical analysis with past work so it's hard to fully understand the significance of their analysis. For example, is the theoretical result similar to past works that work with a fixed dataset or monte-carlo simulation. Similarly, it's unclear if the other methods used in the numerical section achieve similar rates or how it could be substantively different.
2. The numerics section's synthetic experiment provides limited evaluation as the authors could also include results that study the convergence rate of their approach compared to benchmarks. They could also compare different choices of $\phi$.

---

> ### Author Rebuttal · Authors · 2025-07-30
>
> We thank the reviewer for the constructive and thoughtful feedback. For clarity, we have merged related points from the review into grouped responses below.
>
> [**R1Q1**] Convergence rate. The paper does not compare or comment on their theoretical analysis with past work... /  Can any comparison be made about the theoretical convergence rate between the proposed method and other benchmark methods? ... rate presented in this paper compared to the rates SGM, DORO, etc.?  / How does the analysis differ from previous works?
>
> [**R1A1**]
> Thank you for the valuable comment. Under Assumption 1, our method achieves an $\mathcal{O}(1/\sqrt{K})$ convergence rate in iteration complexity, as shown in Theorem 1. This rate is consistent with the rate for the baseline objectives, though derived under distinct assumptions as part of them having distinct objective function. For instance, DORO  [Zhai et al., 2021] achieves the same rate under the assumption that the corresponding objective (line 991) satisfies the same standard convexity and Lipschitz continuity (Assumption 1) and $\chi^2$-DRO can rely on a finite f-divergence constraint and employ mirror descent [Namkoong & Duchi, 2016], achieving the same rate. Optimization with the ERM objective also achieves $\mathcal{O}(1/\sqrt{K})$ without any changes in assumptions.  SGM is the fair baseline, as it has the same CVaR objective. Notably, our method reduces to SGM when the importance weight $w^*$ is a constant (cf. Eq. (7)), yielding the same convergence rate $\mathcal{O}(1/\sqrt{K})$. This equivalence is expected, typically, importance sampling does not improve the asymptotic convergence rate, but rather the constant factor via noise reduction [Zhao & Zhang, 2015].
>
> The key theoretical distinction of our work lies in reducing the stochastic noise floor via pretrained loss-guided sampling:
> We provide a bound on the stochastic noise term $\hat{v}(q)$, which is minimized when sampling from $q(x)\propto w^* (x)p(x)$ where $w^*(x)=(\sqrt{ 2L_{1}\ell(\theta_0;x)}+2L_{2} \kappa)^2 + 1$. Note that this analysis differs from previous works that use importance sampling, as we explicitly connect the importance sampling weight function to the loss value of the given pretrained model. This analysis enables a practical design of $\varphi(\ell)$, and under Assumption (9), the pretrained-loss guided sampling distribution achieves a tighter upper bound on the stochastic noise compared to standard sampling (e.g., SGM).
>
> None of the existing baselines in CVaR optimization or importance sampling provides a stochastic-noise-sensitive analysis that incorporates pretrained models into the sampling distribution.
>
> We will add this discussion and the distinct theoretical contribution in the revised manuscript.
>
>
> ---
>
> [**R1Q2**]  Empirical results that compare different choices of $\varphi$? / How to choose $\varphi$?
>
> [**R1A2**] We thank the reviewer for raising this important question.
>
> The desired form of  $\varphi$ is derived in Eq (7) and given by $\varphi(\ell)=(\sqrt{\ell}+\frac{2L_{2}\kappa}{\sqrt{2L_{1}}})^2 + 1/(2L_1)$ as stated in the Appendix (line 1116). This form arises from matching the optimal importance sampling distribution that minimizes the stochastic noise $\hat{v}(q)$ in Thm 1.
>
> However, in practice, the exact constants $L_{1}, L_{2}$ and $\kappa$ are typically unknown or difficult to estimate. To address this, we propose a practical surrogate $\varphi(\ell) = \ell + c$ where $c$ is a tunable constant. This approximation preserves the dominant order of the loss and offers a simple yet effective implementation.
>
> To empirically study the effect of different $\varphi$, we conducted ablations in Appendix, Section E.2 (Impact of Importance Level Emphasis). We varied $c\in \{0.001, 0.01, 0.1, 1, 10\}$ and evaluated performance across a range of $\beta$ values (quantile). The results show that the method is robust to the choice of $c$, especially at high $\beta$, (e.g., for $\beta=0.99$ the proposed method consistently outperforms all the baselines with all $c$. For $\beta=0.5$, $c\in \{0.001, 0.01, 0.1\}$ lead to performance improvement). It indicates that when large-loss-inducing samples are strongly emphasized, the precise tuning of $\varphi$ becomes less critical.
>
> We agree that this analysis is important and will add this discussion to the main paper in the revised submission for clarity.
>
> ---
> We thank the reviewer again for the valuable feedback. We believe the revised discussion addresses the concern, and we respectfully ask the reviewer to consider increasing their score. We would be happy to provide further clarification if helpful.
>
>
> [Ref-R1-1] Zhao, Peilin, and Tong Zhang. "Stochastic optimization with importance sampling for regularized loss minimization." _International Conference on machine learning_. PMLR, 2015.

---

> ### Author Response · Authors · 2025-08-06
>
> Dear Reviewer,
>
> Thank you again for your valuable and constructive feedback. We hope our detailed responses have addressed your concerns—particularly regarding the convergence rates and the empirical design of the importance weighting scheme. Please let us know if any further clarification or elaboration on these points would be helpful.
>
> Best regards,
>
> the authors

---

### Comment · Area_Chair_FYSe · 2025-08-06

Dear reviewers,

  Please note that reading the authors' rebuttal and discussion are mandatory for NeuRIPS 2025. Can you spend some time to engage with the authors? Thanks!

AC

---

### Author Response · Authors · 2025-08-09

Dear Reviewers and Chairs,

Thank you for your valuable feedback and constructive comments on our manuscript. We greatly appreciate your efforts in reviewing our work.

We are encouraged by the recognition from **all reviewers** of the significance and originality of our main contribution, as reflected by **Reviewer hgR6** _"a straightforward and effective method for improving the efficiency of generating accurate estimates of CVaR"_, **Reviewer dRNj** _"makes sense, is well described, and forms a nice connection between recent advances in generative modeling with guidance and data augmentation techniques for risk-averse optimization", "elegantly designed and well-motivated",_ and **Reviewer uMpj** _"The idea is original, interesting, and can open fruitful avenues both in theory and in practice."_

Moreover, our theoretical analysis was recognized by **Reviewer hgR6** _"provides good motivation for why their approach is effective"_, **Reviewer dRNj**: _"The convergence analysis and ... provide further support and justification for this approach"_, and **Reviewer uMpj**: _"fruitful avenues both in theory..."_. Our numerical analysis was also acknowledged by **Reviewer hgR6**: _"The numerical section while simple highlights intuition for why the proposed approach is effective... Leveraging real-world data also highlights the practical effectiveness of their method."_, and **Reviewer uMpj**: _"The empirical illustrations are well crafted"_.


We also thank the reviewers for their insightful questions. The main points discussed include **[R1A1]** convergence rate comparison, **[R1A2]** empirical choice of $\varphi$, **[R2A1]** additional experiments examining the tradeoff between tail-averse training and standard average performance, **[R2A3]** rationale for using a fixed importance sampling distribution and its extensibility for repeated runs of the proposed algorithm, **[R3A2]** reduced variance in subgradient and iteration loss, and **[R3A5]** application of our method to alternative risk measures (Extremile). We have fully addressed these points in our responses.

We will incorporate all additional results and clarifications into the revised manuscript.

Thank you again.

---

### Decision · Program_Chairs · 2025-09-17

**Decision:**

Accept (spotlight)

**Comment:**

This paper provides a new approach for optimizing conditional value-at-risk (CVaR) over distributions generated from diffusion-based generative models. It proposes an algorithm for reweighting the sampling distribution of the generative diffusion models and theoretically analyze the convergence of their approach. Reviewers liked the algorithm and believed that the idea is original and can offer potentially interesting venues in both theory and practice. The drawback of this paper is that the empirical evaluation is quite limited.

All reviewers suggested acceptance with 2 Accepts and 1 Weak Accept. AC follows consensus to accept the paper.